# Leadership among Women Working to Eradicate Female Genital Mutilation: The Impact of Environmental Change in Transcultural Moments

**DOI:** 10.3390/ijerph17165996

**Published:** 2020-08-18

**Authors:** José Siles-González, Ana Isabel Gutiérrez-García, Carmen Solano-Ruíz

**Affiliations:** Department of Nursing, Faculty of Health Sciences, University of Alicante, 03080 Alicante, Spain; anabel.gutierrez@ua.es (A.I.G.-G.); carmen.solano@ua.es (C.S.-R.)

**Keywords:** female genital mutilation, transcultural nursing, environmental change, women health, cultural moments, critical thinking

## Abstract

The study of cultural moments can identify the level of acceptance of female genital mutilation and the visibility of the involved health problems in a globalized world. Aims: To describe the transcultural process through which immigrant women who have experienced female genital mutilation become leaders against this practice. Method: Descriptive research with cross-cultural principles and a qualitative approach. A semi-structured interview was the chosen technique for data collection. A total of 18 women participated in the preliminary observation and analysis unit, and only 8 women (38.8%) were ideologically against female genital mutilation (FGM). Inclusion criteria: The selected women had undergone FGM and were fully prepared to discuss it. Results: Staying in a different country and the associated social relations reduce cultural pressure and promote critical thinking. Cultural moments reflect the different situations that affect the perception and practice of female genital mutilation. Health problems associated with female genital mutilation (sexual, reproductive, and psychological) become visible at transcultural moments. Conclusions: Environmental country change affects the cultural pressure that sustains this practice in individual minds, institutions, structures, and bodies. These changes produce transcultural moments. The practice of female genital mutilation constitutes a significant segment of gender-based violence.

## 1. Background

This study was part of the ‘Social Challenges Europe 2020 strategy’ program, which considers current political priorities in the field of European strategy, especially focusing on health, demographic change, well-being, and the integration of gender analysis—all of which are inherent within the subject of FGM.

At an international level, Worsley studied perceptions of the little-known practice of female genital mutilation (FGM) [1]. An article studied the social and health risks of FGM, focused on the experiences of women [2]. Further, Ali made references to other stated reasons for FGM, such as social pressure for the maintenance of female virginity, hygiene, and economic explanations (even midwives in the communities of origin advocate this practice for health, moral, and social reasons) [3,4]. Isman and Berggren described the attitudes and changing behavior toward the adoption of a position against this practice in Europe [5]. In Europe, women who are at risk or who have already experienced FGM can use this practice as a basis for seeking asylum. Other studies have focused on analyzing the psychological effects of FGM, which can include post-traumatic stress disorders and memory problems, among others [6]. An article described three cultural moments in which the levels of acculturation or social and communicative integration of women were identified by analyzing the impact of multicultural, intercultural, and transcultural factors in the maintenance, questioning, or abandonment of the ideology of FGM [7]. In Spain, work has been done as part of the theses of Pastor and Jiménez-Ruiz, in order to give women who have experienced FGM a voice and visibility [8,9], as well as to analyze FGM from the perspective of both women and men [9]. Moreover, Ana Silva addressed this problem, in her doctoral thesis, from the perspective of legal and criminal treatment related to FGM in Spain [10].

Approximately 500,000 women who live in the EU have suffered FGM, while 180,000 girls and women are at risk of undergoing it. A map of female genital mutilation in Spain showed 58,200 mutilated women and that some 24,000 girls under 14 are at risk of undergoing FGM. In the province of Alicante, 2000 women are at risk of suffering this practice (8.33% of the total population at risk of FGM in Spain) [11]. The countries of origin are Senegal, which has the largest female population living in Spain (64,000 inhabitants), followed by Nigeria (with 46,000), Mali (with 25,000), and Gambia (with 22,000) [12]. FGM is a practice framed within “harmful traditional practices” (HTP), such as early forced marriage, selective abortion, infanticide, and so on. The World Health Organization (2018) differentiated four types of FGM: Type I (clitoridectomy)—partial or total resection of the clitoris and, only in very rare cases, the foreskin; Type II (excision)—partial or total resection of the clitoris and labia minora, with or without excision of the labia majora; Type 3 (infibulation)—narrowing of the vaginal opening through suturing, stitching, or repositioning of the labia minora and/or labia majora, with or without resection of the clitoris; and Type 4 (other)—any other harmful procedure of the female genitals performed for non-medical purposes, such as punctures, perforations, incisions, scraping, or cauterization of the genital area [12].

The typology (e.g., procedure and timing) of FGM varies according to culture of origin. FGM is a practice that, for the most part, arises from a need to explain the change that girls undergo when they move from childhood to puberty [11,12]. This explanation is done through the narrative or fable that spiritual leaders (e.g., sorcerers, shamans, or priests), imbued with esoteric knowledge, construct by integrating them into belief systems which legitimize patriarchal power [7]. These stories justify FGM for a variety of reasons, such as purity, hygiene, elimination of the male part, safeguarding of virginity, control of women’s sexuality, and so on.

FGM is a process which is very similar to the rites of passage described by Van Geneep as, despite their great variety (due to the diversity of stories on which they are based), they all have in common the institutionalization of the change from childhood to puberty and the systematic control of women’s socialization in the community. This control is manifested in the initiation ceremony where girls, isolated from the rest of the community, are indoctrinated in the current values of femininity [13,14].

The relationship of migrant women with their cultures of origin may remain marked by the values, beliefs, and traditions that uphold FGM as necessary to become accepted and respected women [5]. Even if they are in other countries, there is still great cultural pressure to have their daughters undergo FGM. After passing the initiation test, women are respected by their community and can marry and start a family. Women who do not undergo this process become outcasts who are disowned by their family and community [7]. The complexity of this practice is also reflected in the diversity of its names, which vary from country to country: tradition (Guinea), tisianem (Mauritania), Bundu (Sierra Leone), purification ceremony (Kenya), initiation ceremony (Gambia), and so on [12,14].

Most authors agree that the migration of women does not directly cause acculturation [7,8,9,10,11,14]. Acculturation is a complex process which is directly linked to the potential for communication between migrant women and women in the reception country and their integration into social networks external to those of their countries of origin. It also depends on other factors, such as family situation, cultural level, social class, and work situation, among others [14].

The work of Reig, Siles, and Solano described the fact that health professionals in Spain have insufficient knowledge about FGM [15,16]. Female African leaders created the Inter-African Committee Against Traditional Practices Affecting the Health of Women and Children in 1984, which serves as the basis for global action against FGM. Furthermore, Abusharaf gave a voice to African women in the United States and described the multiple links between the respect and bodily integrity of women, female empowerment, and the economic system [17]. Finally, Hadi analyzed the case of a community of women who had become empowered in the fight against FGM [18], and Thill described the socio-cultural evolution of female genital cutting [19]. We consulted several studies on the incidence of FGM for sexual health problems; for example, Ismail et al. analyzed the incidence of this practice on women’s reproductive health [20] and qualified it as a tragedy for women’s reproductive health [21].

The concepts of “cultural moment”, “habitus”, “logical conformism”, and “technologies of the self” are very useful when analyzing the processes of acculturation. Cultural moments identify the level of acculturation of an individual or a human group from a given culture, depending on their level of communication and interaction with individuals or groups from other cultures. The higher the level of intercultural communication and interaction, the greater the possibility of reflecting on and questioning beliefs and practices about FGM [7]. In a study, three cultural moments were identified, according to the level of communication/interaction: multicultural moments, where there is no communication between people from different cultures, who remain isolated and maintain the beliefs and practices of their countries of origin; intercultural moments, where there is a certain permeability allowing communication between groups from different cultures, but no common objectives or action strategies are established to achieve them collaboratively—this is the phase in which the beliefs and practices of FGM are questioned; and cross-cultural moments, which are characterized by permanent and effective communication between different cultures, allowing the identification of common problems and objectives and the design of collaborative actions to achieve them—in this phase, FGM can be identified as a common problem and objectives and strategies for its eradication can be established [7,22].

For Bordieu, “Habitus” is a “structuring structure” that analyzes the socialization of subjectivity (the process through which social and cultural pressures configure subjectivity) which, from the beginning, is linked to gender, bodies, feelings, and the symbolic power exercised over them [23,24]. It is relevant for studies in which enculturation favors awareness of cultural pressure by making visible some aspects of the social configuration of subjectivity and enabling the questioning of beliefs, values, feelings, and practices that perpetuate male domination [25].

“Logical conformism” is an expression developed by Durkheim to explain how individuals abide by social facts in the face of coercion [26]. Durkheim’s notion of coercion refers to the characterization of social facts as ways of feeling, acting, and thinking which are imposed upon us. Durkheim affirmed that society needs a minimum of logical conformism to exist. This concept is relevant for reflecting on the way in which women adapt to the demands of their social classes and adopt attitudes that are circumscribed to the category to which they belong; not out of obligation, but with the pleasure of feeling that they are fulfilling the social function that corresponds to them in their community [26].

Foucault used the concept of “technologies of the self” to explain the forms of self-configuration which have always existed throughout history, through which individuals are actively constituted in a social and cultural context. It consists of marking the body according to socially established symbolic rules. Hernández Ramírez states that these practices of the technology of self are not devised by the individual but are imposed by models in their own cultures [27]. The practice of FGM is an example of technology of self, where girls are configured as women through a technology of self that marks their bodies such that they can be accepted in their communities.

The Dialectical Structural Model of Care (DSMC) facilitates the dialectical/dynamic vision between categories, such as the functional unit (socializing structures that transmit cultural values, beliefs, practices, and meanings and that could be equivalent to Bordieu’s fields that function externally to the person), the functional element (the receiving individuals that individually and subjectively readjust to the cultural transmission), and, finally, a third category—the functional framework (spaces, settings, or places where FGM practices take place and which, in turn, are linked to culturally transmitted systems of values, beliefs, feelings, and meanings) [28]. This model also makes it possible to assess the impact of an individual’s adaptation or level of resistance to these socializing mechanisms, in which social representation exerts cultural pressure (relations of domination, dependence, and power) on the social interpretation of care.

The aim of this study was to describe the process of gaining awareness and acculturation through which women who have experienced female genital mutilation (FGM) become leaders that advocate against this practice. Our specific objectives were to (a) understand the experiences lived by women who have experienced FGM and who now work toward its eradication; (b) describe the impact of gender on the normalization and invisibility of health problems arising from FGM; (c) identify cultural moments lived by women who have experienced FGM; (d) describe the factors that facilitate or hinder awareness of FGM from the perspective of the dialectical structural model of care; and (e) describe the impacts of “habitus”, “logical conformism”, and “Technologies of the self” in the process of socializing FGM.

## 2. Methods

This exploratory–descriptive research, carried out between April 2018 and November 2019, took a qualitative approach and formed part of a research project which had previously analyzed the experiences of women living in Alicante (Spain) who had experienced FGM. These interviews were originally conducted at the Elche Acoge site (a non-governmental organization) between September 2018 and November 2019 and included women who had been residents in Spain for at least five years and who spoke Spanish; a total of 18 women met these requirements. In this current study, we added a complementary criterion that the women we studied had developed a level of awareness and sensitivity against FGM (i.e., they were female leaders who campaigned against FGM). We took a critical thinking theoretical focus and followed the principles of Habermas’ socio-critical paradigm, which states that people who suffer from a given problem have a voice and are more suited to work on its solution. The socio-critical paradigm was adopted as it promotes studies of practical and emancipatory interest. It is characterized by facilitating people’s participation in the solution of their problems, where communication is a tool for change in practice [29]. Considering its characteristics, a participatory action research (PAR) study was carried out in its first phase (exploratory and reflective) [22,29] to distinguish three types of interest in knowledge: technical interest, which is knowledge of the natural sciences that aims at control and prediction (empirical–analytical sciences); practical interest, which pursues understanding, self-understanding, and the communication of social reality (historical–hermeneutical sciences); and emancipatory interest, which aims to question the prediction and control established by the empirical–analytical sciences (critically oriented sciences) [29].

Durkheim used the concept of “logic conformism” [26] and Bordieu used the concept of “habitus” to promote awareness of the process of socially constructing feelings about FGM [23,24,25]. This study used contributions from the social construction of reality, the world as a representation related to the practice of FGM and interpreting the body and gender as a process of sex construction [30,31,32,33,34].

Context and sample: The Preliminary Observation and Analysis Unit (AU1) comprised 18 women (W1-18) who had experienced FGM. The secondary observation and analysis unit (AU2) comprised seven women leaders (WL1–7) who had all lived through transcultural moments and had become activists advocating against FGM. To identify relevant cultural moments, we used the categories suggested by Siles [7]: intercultural moment—the beginning of communication between different cultures; transcultural moment—change of place becomes cultural change. These were experienced by only seven women, as most women did not communicate with women from other cultures and still thought that FGM is necessary and believe that they should not question something that is so important to their cultures. This group remained under the control of their families (husbands, brothers, or mothers), who did not allow them to relate openly with women from other cultures, nor to have contact with associations that deal with this issue.

We used semi-structured interviews as the data collection technique and the dialectical structural model of care (DSMC) as the data analysis method [28].

In a first session, a semi-structured interview was conducted with open and closed questions, following a script that was given to all the women. The semi-structured interview was chosen due to the difficulties of dealing with a complex subject such as FGM. As a strategy to motivate the women to attend the meeting and to become familiar with the interviewers, they were called for a talk on women’s and children’s health. This interview facilitated the identification of the cultural moments the women were going through. It lasted 3 hours, including the talk. All women answered some common questions, but some were reluctant to answer with regard to some of the issues (those most linked to FGM). Finally, this first interview served as a filter, as women in a multicultural situation were excluded from the next phase.

In the second phase, two more sessions were organized for in-depth individual interviews focused on women who had answered all the questions in a decisive and collaborative way, showing their interest in the FGM theme and who had more potential to communicate with people outside their community of origin (i.e., intercultural moments). The interviewers were two women members of the care culture group. The setting for the interviews was the facility of the NGO “Eche Acoge”. The seven women were interviewed on different days at the Echel Acoge facility. Finally, 14 interviews were conducted (two for each woman). The exploration strategy consisted of obtaining descriptive and structural information (following the guidelines of the MEDC: ideals, scenarios, and personal characteristics), the most significant experiences of the interviewee (as she remembered them), describing specific ideas, feelings, and behaviors around FGM; and three general types of issues. The duration of each focal interview was 1.5 to 2 h. In the course of this interview, the events and processes through which FGM came to be questioned and assessed negatively were identified. The repetition of the same type of problems, significant events, and processes that caused them to reflect on and question FGM were taken as evidence of saturation.

### Ethics Declarations

The Ethics Committee of the University of Alicante (Spain) approved this project on January 2018/Number: UA-2017-12-15. This study is conformed to the ethical principles set out in the Declaration of Helsinki. All the participating individuals signed their informed consent to participation, after having been explained their rights as citizens, the characteristics and objectives of the study, and their guaranteed anonymity.

## 3. Results

### 3.1. Sociological Data:

A total of 18 women participated, with ages ranging from 27 to 61 years (Preliminary Observation and Analysis Unit). Only 38.8% of the women interviewed were ideologically against FGM and were willing to advocate against this practice.

The secondary observation and analysis unit (B) comprised seven women (Table 1):
-WL1 was 46 years old, was born in Guinea Bissau, had lived in Spain for 14 years, was separated, and had a son. She actively participated in an association campaigning against FGM.-WL2 was a native of Kenya who had been living in Spain for 22 years, was separated, and had three children (two boys and one girl). She actively participated in an association campaigning against FGM.-WL3 was the youngest, at 27 years old, who had been born in Guinea Bissau, had lived in Spain for 15 years, was single, and had no children. She actively participated in an association campaigning against FGM.-WL4 was 38 years old, had been born in Mali, was married, had three children (one girl and two boys), and had lived in Spain for 12 years. She participated in several associations campaigning against FGM, but was not collaborating with any particular one. She found it most difficult to fight against FGM actively and attributed this to the care of her children and husband.-WL5 was 61 years old, born in Gambia (capital). She was presently divorced and had lived in Spain since 1974 (45 years). She had five children (three girls and two boys). WL5 participated in several associations campaigning against FGM. She had also become president of a Non-Governmental Organization (NGO).-WL6 was 49 years old, born in Mali. She was divorced and had lived in Spain since 2001. She had two children (two girls). WL6 participated in an association campaigning against FGM.-WL7 was 38 years old, born in Guinea Bissau. She had divorced several years ago. WL7 had three children (two boys and a girl). She had lived in Spain since 2002. She participated in several associations campaigning against FGM but was not collaborating with any particular one.


### 3.2. Female Genital Mutilation and Health Problems

In several African countries, the health systems and their doctors and nurses have become engaged in the practice of FGM, due to its negative health consequences. In Western countries, such as Britain, Sweden, or the United States, it is practiced by physicians under the guise of minimizing the forms of FGM [35]. This poses major ethical and moral problems for doctors and nurses and may confuse the population at risk. Glocalization indicates tendencies toward homogenization (global culture) and heterogenization (local culture) coexisting throughout the modern age.

Although female genital mutilation/cutting (FGM/C) is considered a harmful practice internationally, its practice has been medicalized, especially in the health systems of African countries, supposedly to reduce its negative health effects and, therefore, as a harm reduction strategy in response to these perceived health risks. In many countries where FGM/C is traditionally practiced, prevalence rates of medicalization are increasing, while in countries of migration (e.g., the United Kingdom, the United States of America, or Sweden), trials or repeated statements in favor of alleged minimal forms of FGM/C to replace more invasive forms have raised the debate between medical harm reduction arguments and human rights approaches.

The prevalence or elimination of a practice such as FGM can be interpreted as a dispute between local culture and global culture. The medicalization of FGM in local cultures is the consequence of a synthesis (glocalization) between local and global culture. According to Robertson, the use of the term glocalization means that it is the local culture which assigns meaning to global influences, and therefore, both are interdependent and mutually enabling [36].

Women who have experienced “the rite” (of FGM) and who had become leaders in the advocacy against FGM experience a complex and difficult process. In general, knowledge of FGM alone is not a valid or acceptable way to become a “normal” woman who can be accepted and respected by her community; women must first have access to other cultures and communities in which the role of women is very different to their own experiences. When these women contact strangers from other cultures, they can access different values, beliefs, and norms without having to leave their community of origin. Nonetheless, the process of acculturation usually occurs through the growing phenomenon of emigration to other countries. Indeed, three of the seven women we interviewed in this present study had become acculturated through experiences and contact with women from other cultures. The health problems that arise from FGM are not considered, as such.
(a)Women interpreted difficulties with urination, urinary tract infections, sexual pain, and problems related to penetration as part of normality. They understood that these problems were not usual after they had interacted with women from other cultures:-Four of the women said that they had no desire for sex, due to the pain caused during relationships (W1, W3, W5, W7).-Five women reported that they felt a lot of pain in their sexual relationships (W1, W3, W5, W6, W7) but had thought that the pain was normal.-All women had problems with arousal, lubrication, orgasm, and satisfaction.(b)The women said they had menstrual problems (dysmenorrhea) and genital infection problems:-Six women claimed to have menstrual problems such as dysmenorrhea and dysregulation (W1, W2, W3, W5, W6, W7).-Six women had genital infection problems (W1, W2, W3, W5, W6, W7).(c)Some of the interviewed women had problems during pregnancy and/or delivery (W3 has no sons).-Four women had problems in pregnancy (W1, W4, W5, W6).-Three women had problems in delivery (W3, W5, W6).(d)About the psychological problems associated with FGM.-All women had psychological problems (anxiety, stress, fear).

### 3.3. Female Genital Mutilation and Cultural Moments

Siles and Solano described the influence of multicultural, intercultural, and transcultural factors in the creation of moments which define how women perceive and interpret the ideology of FGM and which may alter their level of acculturation or social and communicative integration [28].

#### 3.3.1. Multicultural Moment: Change of Place without Leaving Cultural Isolation

Women living in this cultural moment experience a barrier between their way of life, their expectations, and their culture. The characteristics of the multicultural moment, in effect, correspond to two situations: (a) women who have lived in Spain only for a short time, and who have not yet left their original circle of reference; and (b) women who, although they have lived in Spain for a long time, are still limited by the influence of their original circle.

In this phase, acculturation that questions the practice of FGM is almost impossible, as these women’s beliefs and identity are linked to the factors above. They cannot socialize within the context of democratic values which project female identity from the perspective of gender equality, as these frameworks identify with beliefs that sustain the practice of FGM as unjust, violent, and criminal. The women we interviewed described how, until they had left their country of origin, it had been impossible to take any other view on this socially rooted practice (Figure 1); for instance, WL1 stated that, in Guinea, “we were all convinced that to be a woman...an honest and respected woman, the tradition was necessary. […] no one would not have agreed with the rite, although that is not talked about either. […] It was just something that did because it had to do”. WL2 said: “In Somalia, in the border area between Ethiopia and Kenya, there was no dissidence regarding any tradition, and it was barely to spoken. Everyone knew that you had to do it to girls, yes, but nothing more.” WL3 stated, “In Guinea Bissau, among the Fulbes nothing was questioned, and everyone agreed.” WL4 said, “In Gambia, there was no cultural pressure to perform the rite. It was something [that was] socially accepted”.

WL7 said: “In Guinea Bissau, there is no problem with Female Genital Mutilation because nobody calls tradition as such. All mothers and grandmothers want their daughters and granddaughters respected in the community”.

#### 3.3.2. Intercultural Moment: The Beginning of Communication between Different Cultures

In this phase, the activation of greater communication and social interaction between different cultures leads women to take a step forward. This situation represents the beginning of a process of recognition and conciliation. New ideas about the roles of women in different cultures are identified. These, in turn, result in an identity transformation. As WL1 put it: “A teacher explained girls’ female anatomy [to us] and what it was for, especially the female genitalia. This experience was at age 12 in 5th grade. Although it was difficult, I asked the teacher. All the students looked at me (the teacher had studied in Russia).

“It was during the beginning of my stay in Cuba that I began to have doubts because women behave differently. […] Then, my trip to Cuba supposed a change in my perspective on these things. In general, my stay in Cuba broadened my mind, but everything was still very difficult.” In this context, WL2 said, “I had a very bad time. I was hardly a girl anymore; I’ll never forget it. I had the infibulation and I had a moment [when I was] tied by the legs, practically without moving for a long time”. WL3 described this phase: “I began to realize that everything we had been told about female genital mutilation was not entirely true when I started travelling to Spain and I was able to talk with other girls.” Similarly, according to WL4, “I started to question things related to female genital mutilation several years after my arrival in Spain because I arrived in 1984, and I began to reflect in 1997”.

WL6 said: “When other women told me that the tradition was bad and dangerous to my health, it was difficult for me to understand. When it was understood, I decided that I had to protect my daughter (my husband was preparing the trip to make them the tradition). Finally, I got a divorce... I suffered a lot during the whole process”.

During intercultural phases, women start to ask themselves, for the first time, questions about the normality of the symptoms caused by their FGM, but they typically do not do anything to resolve these problems as they are immersed in uncertainty and insecurity (Figure 2).

#### 3.3.3. Transcultural Moment: Change of Place Becomes Cultural Change

Cultural socialization occurs as an effect of social interaction and communication between immigrant and native communities, which causes women to start questioning old beliefs. Migrant women become integrated into associations and receive support from professionals and institutions that facilitate these changes of identity within the framework of democratic societies. Critical thinking is one of the pillars of this transformation and involves a change not only of theoretical approaches and beliefs, but also of practices. In line with the socio-critical paradigm [7,28], some of these women end up actively working in the campaign against FGM and the beliefs that underlie it. In the case of WL1, this transcultural moment occurred when she became sufficiently confident to talk about the tradition of FGM with Cuban women, allowing her to interchange and contrast differences in meanings (Figure 3): “In Cuba, [my] friends told me that they had taken something very important from me to feel like a woman and to be able to enjoy my body [...] but what made me change the most was the fact that on my return to Guinea my little sister was being prepared for the rite of initiation. I felt angry and talked with my family, but nobody paid attention to me, and they said that I had gone crazy”.

WL2 said, “When I had my daughter, it was very hard, but I was not willing to have it done to her. It was the beginning because I started talking to people about it and that caused me many problems”.

WL3 stated the following: “We were on holiday in the Pyrenees in July 2005. We were camping with my Spanish father, and there I caught an infection in the genital area. He took me to a medical center and a gynecologyst or doctor; I do not remember, did an exploration. Afterwards, he called a co-worker (something was out of the ordinary). Later they spoke with my father, who until that day, did not know that female genital mutilation (clitoridectomy) had been practiced on me, and it was that day that I discovered that what in my culture was special and was celebrated with gifts, was the opposite, and then I started to live it [my reality] differently.”

WL4 recalled, “I contacted people who made me question the ritual in 1997 (I had arrived in Spain in 1984). I am going to write an autobiographical book in which I will talk about all this, and it will help me to reflect [upon it].”

In this transcultural phase, these seven women had been considered traitors by their community and family because they had reacted against FGM. However, they overcame their uncertainty and, in parallel with the fight against FGM, identified their health problems and were able to link them to this practice.

### 3.4. Female Genital Mutilation from the Perspective of the Dialectical Structural Model of Care

Functional unit: The feelings, beliefs, values, traditions, and norms (FU) that favor the practice of FGM [26] are not exclusively religious. As WL1 stated: “My father is Muslim, but my mother is Catholic, so there was mixing, and the practice transcended the purely religious issue [...] There are many ethnicities and each one has an [different] influence and also looks for different names to insult women who do not want tradition.” Rather, a series of transversal values (not exclusively religious ones) support this practice, as also described by WL1: “Tradition ensures the purity of women, hygiene, and even femininity; a woman without tradition is not a total woman.” Similarly, WL2 said, “Religion influences the rite, but this tradition does not come from the Koran. It is a tradition that must be done, and that is that [...] The Masahi are Christians and they also have this tradition [...].” WL3 put it this way:

“For the Fulbe culture, my community, it is unthinkable not to do this practice. Female genital mutilation is not exclusive of Islam [or] established in the Koran but is a tradition so ingrained that it is unacceptable not to continue doing it”.

WL4 described the following: “The community shares the belief that female genital mutilation must be carried out. There is no resistance because the people around [there] do not think about eradicating it. There is no pressure because there is no capacity to decide; there is no need for pressure. This situation is as much for family education as [it is] for street [education]”.

Functional framework: The functional framework (FF) is constituted of the scenarios where FGM occurs [26]. The tradition of FGM is often carried out in places which are especially prepared and separated from the community; usually in a hut away from the others or even outdoors. It is also sometimes performed in the field near some type of symbol, such as a sacred tree (as in the cases of WL1, WL2, and W5). However, in some cases, FGM is practiced in the family home, usually in the kitchen (as for WL3). There is also an intermediate place (separated from the community but not where the practice is carried out) where the process of socialization takes place after the mutilation. As affirmed by WL1, “It can last several days, and in its course, the girls indoctrinated about the role of women that they should perform in the future.” WL4 decided not to talk about where or how FGM was practiced on her. The seven women all spoke with great affection for their mothers and grandmothers, but they had all had confrontations with them. These intergenerational confrontations had lasted for a long time, but, eventually, they had all spoken with these relatives and had forgiven them.

Functional element: The functional element (FE) is constituted by the people who practice FGM and also those who are victims of this practice [26]. Older women in these societies (sometimes called “slicers”) or, in many cases, family grandmothers usually carry out the tradition of FGM; the community highly respects both. As WL1 confirmed, “Because deep down, the grandmothers are the ones that are going to get the girls to integrate into the community, and they want both the girls and their mothers.” WL2 described her FGM thus: “At five years old at my grandmother’s house. A cutter was the one that did the practice”. For WL3, it was at “Almost five years old. Older women were the ones who [did] the female genital mutilation in a field”. WL4 said, “I think it was an old cutter dedicated to these things”. More recently, the tradition has started to become medically institutionalized, with doctors and nurses starting to perform the practice; according to WL1, “Now doctors also perform the tradition for health reasons, because many girls have died” [35].

## 4. Discussion

In addition to describing the process by which women who have undergone FGM become leaders, another contribution of this research is to describe the link between women’s transcultural experiences (sensitized against the practice of FGM) and the awareness (visibility) of these women about their health problems (urological, sexual, reproductive, and psychological). Other studies have described the psychological effects of FGM, such as post-traumatic stress disorders and memory problems, among others [7]. One article described three cultural moments, in which women’s levels of acculturation or social and communicative integration were identified by analyzing the impact of multicultural, intercultural, and transcultural factors in the maintenance, questioning, or abandonment of the ideology of FGM [7].

Regarding the dynamics and changes experienced in cultural moments, the women in this study accepted, within their multicultural moment, FGM as a beneficial “tradition”. They felt that, in order to become women and integrate into their communities, they had to comply with the rite of passage represented by FGM [13].

They believed in its positive effect on hygiene, health, sex, and reproduction; in other words, the peculiarities of logical conformism and habitus provided by Durkheim [26] and Bordieu to understand the phenomena of socialized subjectivity and the effects of cultural pressure on these women in their communities of origin [23,24,25]. However, migration and intercultural and transcultural contact (communication as a tool for change in practice) allowed these women to reflect on FGM and interpret themselves from other feminine models which allow women to fulfil themselves outside FGM, both in Spain and in other Western countries [7,8,9,10].

In contrast, women’s continued adhesion to FGM is greater when they remain in their countries of origin (e.g., Sudan, Ethiopia, Mali, Guinea Bissau, and so on), where they have more difficulty in leaving the multicultural moment [3,4,11].

None of the women who arrived at the transcultural moment continued to believe that FGM has health benefits [5,7]. The results of this study confirm the findings of Abdalla and Belay, Abdalla and Galea Tiruneh, and Kia: Women have health, psychological, and social problems associated with FGM [2,6,21], along with problems relating to sexual desire and arousal, pain with penetration, and lubrication. This study also confirms the existence of problems in women’s reproductive health [20,21].

Women consider FGM to be related to the social construction of gender and the representation of the body [32,33]. All women say that FGM has been imported due to globalization [34,36,37]. Women living in the transcultural moment have become aware of the physical and psychological violence implied by FGM, which is exercised by symbolic power and has been legitimized by patriarchal political systems [27,34,38,39].

We do not know whether FGM was a cause or a consequence of the values, beliefs, or norms of the seven women interviewed in this study. However, they do not doubt the technology that marks their bodies and reinforces their adhesion to these factors. This body mark also highlights their identity within the category of being female [27]. These women have experienced the social and historical construction of gender as dynamic processes in different and conflicting contexts, with respect to the practice of FGM [33,34].

FGM affects all these planes. Likewise, studies by various authors have endorsed changes in these ways of thinking and of acting concerning tradition, after immigrants who have been under the influence of other cultures for a long time undergo the processes of acculturation and cultural change after migration [30,31].

Divorced women said they had limited communication outside of the circle marked by their husbands until their separation. As a result of their divorces, the women became more conscious of the impact of FGM on the construction of female identity, in the context of dependence on masculine power [7]. In accordance with Durkheim’s logical conformism and Bordieu’s habitus, which promotes symbolic power and masculine [23,24,25], women internalized in their subjectivity the norms which regulate the feminine in their community of origin. Women thought that this dependence on men was normal and that FGM was good, because their husbands loved them with the practice carried out [30,31,32,33,34].

This habitus uses the social construction of the body as a tool for forming structures related to female identity, where socialized subjectivity (i.e., interpersonal and intrapersonal interactions) plays a determining role in developing this construction. Male power has long been creating norms, values, and symbols that nourish argumentation for the cultural construction of gender, enhancing an ideal and exemplary female identity [12,13]. Before this masculine power can act on the ideology or conscience of Foucault, a physical technology of the body such as FGM must already exist [27].

This situation contributes to the subsequent subjective and intersubjective assimilation of a corollary of beliefs, norms, and values which extol FGM as an essential bulwark from which the ideal model of a woman is built [13].

### Limits of Study

Problems in identifying groups of migrant women who have experienced FGM.

Need for collaboration with intermediary institutions (NGOs, social services, associations) to connect with groups of migrant women who have experienced FGM.

The difficulties of communicating about FGM with migrant women who are in a multicultural moment.

FGM is a very complex issue which provokes fear and suspicion among migrant women who are in a multicultural moment.

Women who are in a transcultural situation have suffered the consequences of their change of attitude toward FGM: rejection by their husbands, disaffection from their families, marginalization in their communities of origin, and uprooting from migrant groups in a multicultural situation.

## 5. Conclusions

Change of location can lead to a decrease in cultural pressure (the cultural pressure that facilitated FGM).

As a consequence, moving to a new country can encourage critical thinking and cross-cultural perspectives on the practice of FGM.

The practice of FGM constitutes a significant segment of gender violence, which leads to both physiological and psychological health problems. Factors such as religion and hygiene or the degree of women’s purity affect the practice of FGM, but the mechanisms of cultural pressure established for the continuation of this practice (cultural determinants) perpetuate it. The process of acculturation that occurs when women confront other cultures is the essential basis for them to discuss FGM. Acculturation also influences the development of critical thinking and promotes cross-cultural moments.

The concepts of “cultural moment”, “habitus”, “logical conformism”, and “technologies of the self” are very useful for analyzing the processes of acculturation involved in FGM practices.

The DSMC facilitated the relational analysis of the ideals (beliefs, stories, traditions, myths), personal characteristics, and scenarios of the people directly involved in the practice of FGM (girls, mothers, grandmothers, cutters, and community leaders). It is very pertinent to value the cultural moments and the evolution of the situation of emigrant women from the perspectives of logical conformism and habitus. The DSMC identifies levels of acculturation, cultural moments, and the existence of reflection/questioning about FGM, such as ideas and beliefs, family and work situations, cultural levels, social class, and economic situation.

Intergenerational differences between women who have or have not experienced FGM only appear alongside transcultural moments. Women after acculturation—a cross-cultural moment—may even become activists and campaigners in fighting FGM (as in the case of the seven women we interviewed in this study).

Health problems associated with female genital mutilation (sexual, reproductive, and psychological) become visible at transcultural moments. Therefore, we argue that critical thinking among these women is only possible in situations of transculturality and glocalization.

The implications for practice and key phases for program development, implementation, and evaluation from the perspective of the PAR:-Facilitate the empowerment of women by enhancing their integration into associations that support the eradication of FGM (all are working with different associations);-Transform women into educational agents to raise awareness among migrant women about the reality of FGM. Several programs have already been carried out by women participants in different settings;-Transform women participants into educational agents to make women in their countries of origin aware of the reality of FGM. A project has already been implemented jointly by WL and members of the group Culture of Care (program of awareness activities against female genital mutilation developed in Guinea Bissau in December 2019);-Sensitize health professionals through awareness courses on FGM (several courses have already been implemented in different health centers);-Include programs to raise women’s awareness of the reality of FGM, implemented in associations such as Elche Acoge and other NGOs.

Key phases for elaboration, implementation, and evaluation of programs from a PAR perspective are:-Phases of the PAR: identify the problem (or DX); design of action plan; execution of the action plan; observation, collection, and analysis of information; reflection and reinterpretation of results; re-planning and re-evaluation of the problem.

## Figures and Tables

**Figure 1 ijerph-17-05996-f001:**
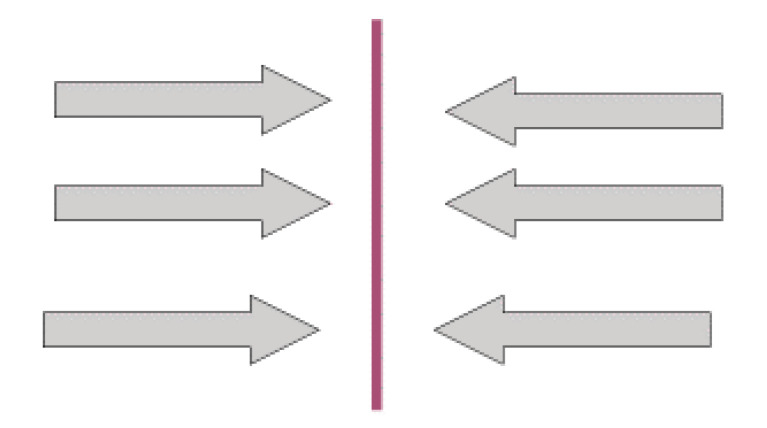
Cultural moments: female genital mutilation (FGM) and Multiculturalism. Source: Siles-González, J.; Reig-Alcaraz, M.; Noreña, A.L.; Solano-Ruiz, C. [7].

**Figure 2 ijerph-17-05996-f002:**
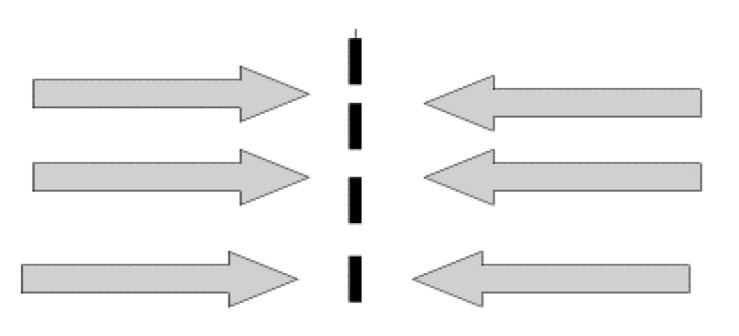
Cultural moments: FGM and Interculturalism. Source: Siles-González, J.; Reig-Alcaraz, M.; Noreña, A.L.; Solano-Ruiz, C. [7].

**Figure 3 ijerph-17-05996-f003:**
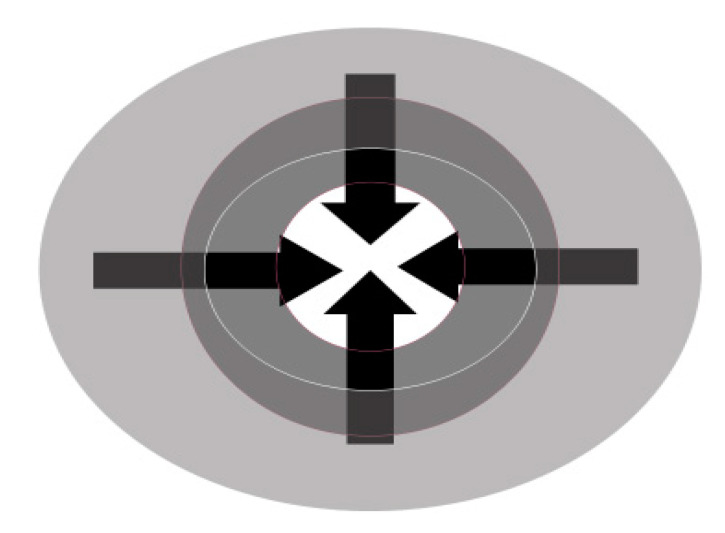
Cultural moments: FGM and Transculturalism. Source: Siles-González, J.; Reig-Alcaraz, M.; Noreña, A.L.; Solano-Ruiz, C. [7].

**Table 1 ijerph-17-05996-t001:** Women who have lived the transcultural moment.

Women Leaders	Age	Country of Origin	Time in Spain (years)	Civil Status Activity	Children	Activity Against the MGF
WL1	46	Guinea Bissau	14	Separate	1 girl	Active partition in associations
WL2	53	Kenya	22	Separate	3 (2 boys and 1 girl)	Active partition in associations
WL3	27	Guinea Bissau	15	Single	No	Active partition in associations
WL4	38	Mali	12	Married	3 (2 boys and 1 girl)	Occasional collaborator
WL5	61	Gambia	44	Divorced	5 (3 girls and 2 boys)	Active partition in associations (President of an association against FGM)
WL6	49	Mali	18	Divorced	2 girls	Active partition in associations
WL7	38	Guinea Bissau	17	Divorced	3 (2 boys and 1 girl)	Occasional collaborator

Source: Author’s elaboration based on the interviews carried out.

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
