# Peer review of "Leadership among Women Working to Eradicate Female Genital Mutilation: The Impact of Environmental Change in Transcultural Moments"

_ijerph, 2020, doi:10.3390/ijerph17165996_

Round 1
Reviewer 1 Report
This is an interesting piece of work that addresses the awareness and acculturation of women who have undergone female genital mutilation (FGM), leaders against this practise. Title: revise the expression “working to working to”, this might be a mistake. Background: start with a general objective, followed by specific objectives. Authors must introduce FMG, incidence in Europe, Spain, origin of women, type of FGM and how and when it is performed, relationship with culture of origin, migration and acculturation in recipient countries. Concepts like “cultural moments” “dialectical structural model of care” "habitus", "logical conformism" and "technologies of the self” should be defined in this section for the reader to better understand. Authors provide a bibliography but it should be structured according to a plan, ending with the change in these women and the need to explore it. Finish this section with the objective of the investigation. Methods: qualitative study in the critical theory framework. A reference to the socio-critical paradigm and the Habermasian concept of emancipatory interest is necessary (Habermas J. (2005) Conocimiento e interés. En: Habermas J. Ciencia y técnica como ideología. Madrid, Tecnos). Empowerment, praxis and values are part of participatory action research (PAR). Researchers could put their study in the first phase (initial idea/exploration/reflection) of PAR. Context and sample: describe “cross-cultural moments” lived by only 7 women. Why did the rest not experience it? What was the duration of the interviews? Describe the setting better. Did you use the interview script? Was there data saturation? Results: sociodemographic data of the participants (3.1) would be more visible in a table. Authors describe well the health problems linked to FGM (3.2). In qualitative research they should be supported with quotes from participants rather than quantify responses. Sections 3.3 and 3.4 show interesting results. Dialectical structural model of care should be explained in the introduction/background. More information about doctors and nurses performing this practise is necessary. Discussion: What are the authors referring to with this expression: ‘they do not doubt that the technology that 290 marks their bodies and reinforces their adhesion to these factors’ (p. 7, 290-291)? A more focused discussion on advocacy and women’s empowerment against FCM is needed. The last paragraph of the discussion should be moved to the beginning, following the order of the results. Conclusions: well presented, responding to the objectives. It is missing implications for practise and keys steps to implement evaluable programmes from a PAR perspective.
Author Response
Response to Reviewer 1 Comments
(These answers and the manuscript are pending the English language edition)
Point 1: Title: revise the expression “working to working to”, this might be a mistake.
Response 1: Revised title: Leadership among Women Working to eradicate Female Genital Mutilation: The Impact of Environmental Change in Transcultural Moments
Point 2: Background: start with a general objective, followed by specific objectives. Authors must introduce FMG, incidence in Europe, Spain, origin of women, type of FGM and how and when it is performed, relationship with culture of origin, migration and acculturation in recipient countries.
Concepts like “cultural moments” “dialectical structural model of care” "habitus", "logical conformism" and "technologies of the self” should be defined in this section for the reader to better understand. Authors provide a bibliography but it should be structured according to a plan, ending with the change in these women and the need to explore it. Finish this section with the objective of the investigation.
Response 2: Background
- Approximately 500,000 women who live in the EU have suffered FGM, and 180,000 girls and women are at risk of undergoing it. The map of female genital mutilation in Spain shows 58,200 mutilated women and some 24,000 girls under 14 are at risk of undergoing FGM. In the province of Alicante 2000 women are at risk of suffering this practice (8.33% of the total population at risk of FGM in Spain). The countries of origin are Senegal, which has the largest female population living in Spain (64,000 inhabitants, followed by Nigeria with 46,000, Mali with 25,000 and the Gambia with 22,000 [12]. FGM is a practice framed within "harmful traditional practices" (HTP), such as early forced marriage, selective abortion, infanticide,
- World Health Organization (2018) differentiates four types of FGM: Type I: Clitoridectomy: partial or total resection of the clitoris and, only in very rare cases, the Type II: Excision: partial or total resection of the clitoris and labia minora, with or without excision of the labia majora Type 3: Infibulation: narrowing of the vaginal opening through suturing, stitching or repositioning of the labia minora and/or labia majora, with or without resection of the clitoris Type 4: Other: any other harmful procedure of the female genitals performed for non-medical purposes, such as punctures, perforations, incisions, scraping or cauterization of the genital area [13].
- Typology such as procedure and timing of FGM varies according to culture of FGM is a practice that, for the most part, arises from the need to explain the change that girls undergo when they move from childhood to puberty. This explanation is done through the narrative or fable that spiritual leaders (sorcerers, shamans, priests), imbued with esoteric knowledge, construct by integrating them into belief systems that legitimize patriarchal power. These stories justify FGM for a variety of reasons: purity, hygiene, elimination of the male part, safeguarding of virginity, control of women's sexuality, etc.
FGM is a process very similar to rites of passage described by Van Geneep because despite their great variety (because of the diversity of stories on which they are based) they all have
in common the institutionalization of the change from childhood to puberty and the systematic control of women's socialization in the community. This control is manifested in the initiation ceremony where girls, isolated from the rest of the community, are indoctrinated in the current values of femininity [14].
- The relationship of migrant women with their cultures of origin is still marked by the values, beliefs and traditions that uphold FGM as necessary to become accepted and respected women. Even if they are in other countries there is great cultural pressure to have their daughters undergo After passing the initiation test, women are respected by their community and can marry and start a family. Women who do not undergo this process become outcasts who are disowned by their family and community [8]. The complexity of this practice is also reflected in the diversity of its names, which vary from country to country: tradition (Guinea), tisianem (Mauritania), Bundu (Sierra Leone), purification ceremony (Kenya), initiation ceremony (Gambia), etc. [14].
Most authors agree that women's migration does not directly cause acculturation [8, 9, 10, 12]. Acculturation is a complex process that is directly linked to the potential for communication between migrant women and women in the reception country and their integration into social networks external to those of their countries of origin. It also depends on factors such as: family situation, cultural level, social class, work situation, etc. [15].
- The concepts of "cultural moment", "habitus", "logical conformism" and "technologies of the self" are very useful for analysing the processes of acculturation. Cultural moments identify the level of acculturation of an individual or a human group from a given culture depending on their level of communication and interaction with individuals or groups from other cultures. The higher the level of intercultural communication and interaction, the greater the possibility of reflecting on and questioning beliefs and practices about FGM [8]. In a study, three cultural moments are identified according to the level of communication/interaction: Multicultural moment: there is no communication between people from different cultures, but they remain isolated maintaining all the beliefs and practices of their countries of origin; Intercultural moment: there is a certain permeability allowing communication between groups from different cultures, but no common objectives or action strategies are yet established to achieve them collaboratively. It is the phase in which the beliefs and practice of the FGM are questioned; Cross-cultural moment: it is characterized by a permanent and effective communication between different cultures that allows the identification of common problems and objectives and the design of collaborative actions to achieve them. In this phase, the GFM can be identified as a common problem and objectives and strategies for its eradication can be established [8].
The Dialectical Structural Model of Care (DSMC) facilitates the dialectical/dynamic vision between categories such as: the functional unit (socializing structures that transmit cultural values, beliefs, practices and meanings and that could be equivalent to Bordieu's fields that function externally to the person), the functional element (the receiving persons that individually and subjectively readjust the cultural transmission) and, finally, a third category: the functional framework (spaces, settings or places where FGM practices take place and which, in turn, are linked to culturally transmitted systems of values, beliefs, feelings and meanings). This model also makes it possible to assess the impact of the individual's adaptation or level of resistance to these socializing mechanisms, in which social representation exerts cultural pressure (relations of domination, dependence and power) on the social interpretation of care.
For Bordieu, "Habitus" is a "structuring structure" that analyzes the socialization of subjectivity (the process through which social and cultural pressure configure subjectivity), which from the beginning is linked to gender, bodies, feelings and the symbolic power exercised over them [25, 26]. It is relevant for studies in which enculturation favors awareness of cultural pressure by making visible some aspects of the social configuration of subjectivity and the questioning of beliefs, values, feelings and practices that perpetuate male domination [27].
"Logical conformism" is an expression developed by Durkheim to explain how individuals abide by social facts in the face of coercion. Durkheim's notion of coercion refers to the characterization of social facts as ways of feeling, acting and thinking that are imposed on us. Durkheim affirms that society needs a minimum of logical conformism to exist. This concept is relevant for reflecting on the way in which women adapt to the demands of their social classes and adopt attitudes that are circumscribed to the category to which they belong, not out of obligation, but with the pleasure of feeling that they are fulfilling the social function that corresponds to them in their community.
24] Foucault uses the concept of "technologies of the self" to explain the forms of self- configuration, which have always existed throughout history, through which individuals are actively constituted in a social and cultural context. It consists of marking the body according to socially established symbolic rules. Hernández Ramírez states that these practices of the technology of self are not devised by the individual, but are imposed by models in their own cultures. [28] The practice of the GFM is an example of technologies of self where girls are configured as women through a technology of self that marks their bodies so that they can be accepted in their communities.
- Bibliography was structured according to a
- Background finish with the objective of the
Point 3: Methods: qualitative study in the critical theory framework. A reference to the socio- critical paradigm and the Habermasian concept of emancipatory interest is necessary (Habermas J. (2005) Conocimiento e interés. En: Habermas J. Ciencia y técnica como ideología. Madrid, Tecnos). Empowerment, praxis and values are part of participatory action research (PAR). Researchers could put their study in the first phase (initial idea/exploration/reflection) of PAR.
Response 3: Method
1.-we adopted the socio-critical paradigm because it promotes studies of practical and emancipatory interest. It is characterized by facilitating people's participation in the solution of their problems and the communication is a tool for change in practice. Considering its characteristics, a participatory action research (PAR) study has been carried out in its first phase (exploratory and reflective) [23]. Habermas [29] distinguishes three types of interest in knowledge: Technical interest: knowledge of the natural sciences that aims at control and prediction (empirical-analytical sciences); practical interest that pursues understanding, self- understanding and communication of social reality (historical hermeneutical sciences); and emancipatory interest that aims to question the prediction and control established by the empirical-analytical sciences (critically oriented sciences).
Point 4 Context and sample: describe “cross-cultural moments” lived by only 7 women. Why did the rest not experience it? What was the duration of the interviews? Describe the setting better. Did you use the interview script? Was there data saturation? Results: sociodemographic data of the participants (3.1) would be more visible in a table.
Response 3: Context and sample
Intercultural and transcultural moments are described in (pp 11-13) Intercultural moment: the beginning of communication between different cultures and Transcultural moment: Change of place becomes cultural change. They are experienced by only 7 women because most women do not communicate with women from other cultures and still think that the GFM is necessary and believe that they should not question something that is so important to their cultures. Also, this group remains under the control of their families (husbands, brothers or mothers) who do not allow them to relate openly with women from other cultures, nor have contacts with associations that deal with this issue.
Point 4 What was the duration of the interviews? Describe the setting better. Did you use the interview script? Was there data saturation
Response 4: In a first session, a semi-structured interview was conducted with open and closed questions following a script that was given to all the women. The semi-structured interview was chosen because of the difficulties of dealing with a complex subject such as the FGM. As a strategy to motivate the women to attend the meeting and to become familiar with the interviewers, they were called for a talk on women's and children's health. This interview facilitated the identification of the cultural moments the women were going through. It lasted 3 hours, including the talk. All women answered some common questions but some were reluctant to answer some of the issues (those most linked to the GFM). Finally, this first interview served as a filter since women in a multicultural situation were excluded for the next phase.
In a second phase, two more sessions were organized for in-depth individual interviews focused on women who had answered all the questions in a decisive and collaborative way, showing their interest in the GFM theme and had more potential to communicate with people outside their community of origin (intercultural moment). The interviewers were 2 women members of the care culture group. The setting for the interviews was the facilities of the NGO "Eche Acoge". The seven women were interviewed on different days at the Echel Acoge facility. Finally, 14 interviews were conducted (2 to each woman).
The exploration strategy consisted in obtaining descriptive and structural information (following the guidelines of the MEDC: ideals, scenarios, and personal characteristics), the most significant experiences of the interviewee as she remembered them, describing specific ideas, feelings, and behaviors around the FGM. three general types of issues: The duration of each focal interview was 1.30 to 2 hours. In the course of this interview, the events and processes through which the MGF came to be questioned and assessed negatively were identified. The repetition of the same type of problems, significant events and processes that caused them to reflect on and question the FGM was evidence of saturation.
Point 5 Results: sociodemographic data of the participants (3.1) would be more visible in a table. In qualitative research they should be supported with quotes from participants rather than quantify responses. Sections 3.3 and 3.4 show interesting results. More information about doctors and nurses performing this practise is necessary.
Response 5:
- We elaborated a table with sociodemographic data (see revised manuscript)
- The study was supported by quotes from the participants
- About medicalization of FGM:
In several african countries the health systems and their doctors and nurses, have engaged in the practice of GFM because of the negative consequences for health. In Western countries such as Britain, Sweden or the United States, it is practiced by physicians under the guise of minimizing the forms of GFM. This poses major ethical and moral problems for doctors and nurses and may confuse the population at risk [35] .
Although female genital mutilation/cutting (FGM/C) is internationally considered a harmful practice, its practice is being medicalised especially in the health systems of African countries supposedly to reduce its negative health effects and is therefore suggested as a harm reduction strategy in response to these perceived health risks. In many countries where FGM/C is traditionally practiced, prevalence rates of medicalization are increasing, and in countries of migration, such as the United Kingdom, the United States of America or Sweden, trials or repeated statements in favour of alleged minimal forms of FGM/C to replace more invasive forms, have raised the debate between medical harm reduction arguments and a human rights approach. The prevalence or elimination of a practice such as GMF can be interpreted as a dispute between local culture and global culture. The medicalization of the GFM in local cultures is the consequence of a synthesis (glocalization) between local culture and global culture. According to Robertson, the use of the term glocalization means that it is local culture which assigns meaning to global influences, and therefore both are interdependent and mutually enabling [36].
Point 6 Discusión: What are the authors referring to with this expression: ‘they do not doubt that the technology that marks their bodies and reinforces their adhesion to these factors’; A more focused discussion on advocacy and women’s empowerment against FCM is needed. The last paragraph of the discussion should be moved to the beginning, following the order of the results
Response 6:
6.1 They do not doubt that this technology marks their bodies and reinforces their adhesion to the social construction of sex and gender through body technologies (highly symbolic body modifications) [27, 32, 33]
6.2. A more focused discussion has developed on the advocacy and empowerment of women against the FCM (see revised manuscript)
6.3 We moved the last paragraph of the discussion to the beginning.
Point 7; implications for practise and keys steps to implement evaluable programmes from a PAR perspective.
Response 7: We write the implications for practice and key steps for implementing programs that can be evaluated from an PAR perspective.
-Facilitate the empowerment of women by enhancing their integration into associations that support the eradication of the GFM (all are working with different associations).
-Transforming women into educational agents to raise awareness among migrant women about the reality of the GFM. Several programmes have already been carried out by women participants in different settings.
-Transforming women participants into educational agents to make women in their countries of origin aware of the reality of the GFM. A project has already been implemented jointly by WL and members of the group Culture of Care (program of awareness activities against female genital mutilation developed in Guinea Bissau in December 2019).
-Sensitizing health professionals through awareness courses on FGM. (several courses have already been implemented in different health centres).
-Programs to raise women's awareness of the reality of the GFM implemented in associations such as Elche Acoge and other NGOs.
Key phases for elaboration, implementation and evaluation of programs from a RAP perspective
-Phases of the PAR: identify the problem, design of action plan, execution of the action plan, observation, collection and analysis of information, reflection, reinterpretation of results, re- planning, re-evaluation of the problem.

Reviewer 2 Report
I congratulate the authors for the object of study. I believe that it will contribute a lot to science, however, it needs adjustments.
The introduction has very old references and will need updating. Opt for reviews under 5 years old. What is the purpose of the study? This must be inserted at the end of the introduction.
The method needs a better description. It is fragile. How was the study studied? Better describe the data collection process. And the bioethical aspects? Which theory supports this qualitative research? Is it a local study?
In line 93, results and discussion were written. According to the magazine's rules, these elements are separated. Please remove the discussion name.
The discussion needs depth. The results were not compared with the area and international literature. What has been achieved with the results? What are the contributions to the area? What are the limits?
The conclusion must be objective and respond to the objectives of the study. Need modification.
References need updates.
This is my opinion.
Author Response
Response to Reviewer 2 Comments
(The manuscript is pending review english language)
Point 1: The introduction has very old references and will need updating. Opt for reviews under 5 years old. What is the purpose of the study? This must be inserted at the end of the introduction.
Response 1:
- The references in introduction was updated
- The purpose of the study has been inserted at the end of the introduction.
Point 2: The method needs a better description. It is fragile. How was the study studied? Better describe the data collection process. And the bioethical aspects? Which theory supports this qualitative research? Is it a local study?
Response 2:
2.1 In a first session, a semi-structured interview was conducted with open and closed questions following a script that was given to all the women. The semi-structured interview was chosen because of the difficulties of dealing with a complex subject such as the FGM. As a strategy to motivate the women to attend the meeting and to become familiar with the interviewers, they were called for a talk on women's and children's health. This interview facilitated the identification of the cultural moments the women were going through. It lasted 3 hours, including the talk. All women answered some common questions but some were reluctant to answer some of the issues (those most linked to the GFM). Finally, this first interview served as a filter since women in a multicultural situation were excluded for the next phase.
In a second phase, two more sessions were organized for in-depth individual interviews focused on women who had answered all the questions in a decisive and collaborative way, showing their interest in the GFM theme and had more potential to communicate with people outside their community of origin (intercultural moment). The interviewers were 2 women members of the care culture group. The setting for the interviews was the facilities of the NGO "Eche Acoge". The seven women were interviewed on different days at the Echel Acoge facility. Finally, 14 interviews were conducted (2 to each woman). The exploration strategy consisted in obtaining descriptive and structural information (following the guidelines of the MEDC: ideals, scenarios, and personal characteristics), the most significant experiences of the interviewee as she remembered them, describing specific ideas, feelings, and behaviors around the FGM. three general types of issues: The duration of each focal interview was 1.30 to 2 hours. In the course of this interview, the events and processes through which the MGF came to be questioned and assessed negatively were identified. The repetition of the same type of problems, significant events and processes that caused them to reflect on and question the FGM was evidence of saturation.
2.2 The Ethics Committee of the University of Alicante (Spain) approved this project on January 2018/Number: UA-2017-12-15.This study is conformed to the ethical principles set out in the Declaration of Helsinki. All the participating individuals signed their informed consent to participation, after having been explained their rights as citizens, the characteristics and objectives of the study, and their guaranteed anonymity.
2.3 We took a critical thinking theoretical focus and followed the principles of Habermas’ sociocritical paradigm which states that people who suffer from a problem have a voice and are more suited to work on its solution. The socio-critical paradigm has been adopted because it promotes studies of practical and emancipatory interest. It is characterized by facilitating people's participation in the solution of their problems and the communication is a tool for change in practice[29]. Considering its characteristics, a participatory action research (PAR) study has been carried out in its first phase (exploratory and reflective) [22, 29] distinguishes three types of interest in knowledge: Technical interest: knowledge of the natural sciences that aims at control and prediction (empirical-analytical sciences); practical interest that pursues understanding, self-understanding and communication of social reality (historical hermeneutical sciences); and emancipatory interest that aims to question the prediction and control established by the empirical-analytical sciences (critically oriented sciences)[29].
Durkheim used the concept of ‘logic conformism’[27], and Bordieu used the concept of “habitus” to promote awareness of the process of socially constructing feelings about FGM [23, 24, 25]. This study used contributions from the social construction of reality, the world as representation related to the practice of FGM and interpreting the body and gender as a process of sex construction [30, 31, 32, 33, 34].
2.4 This is a local study with women with residential instability (they often change their place of residence to different provinces).
Point 3: Results and discussion were written. According to the magazine's rules, these elements are separated. Please remove the discussion name.
Response 3: Discussion deleted
Point 4: The discussion needs depth. The results were not compared with the area and international literature. What has been achieved with the results? What are the contributions to the area? What are the limits?
Response 4:
4.1 Discussion modified: Comparison of results with literature from the national and international contex.
4.2 Increase the knowledge about the GFM in the studied context. -Facilitate the empowerment of women by enhancing their integration into associations that support the eradication of the GFM (all are working with different associations).
-Transforming women into educational agents to raise awareness among migrant women about the reality of the GFM. Several programmes have already been carried out by women participants in different settings.
-Transforming women participants into educational agents to make women in their countries of origin aware of the reality of the GFM. A project has already been implemented jointly by WL and members of the group Culture of Care (program of awareness activities against female genital mutilation developed in Guinea Bissau in December 2019).
-Sensitizing health professionals through awareness courses on FGM. (several courses have already been implemented in different health centres).
-Programs to raise women's awareness of the reality of the GFM implemented in associations such as Elche Acoge and other NGOs.
4.3 The essential contribution in the area is to identify the dynamic and complex reality of migrant women who have experienced the GFM. Likewise, to describe the process of acculturation from the perspective of cultural moments and to evidence the importance of communication as a tool of change in the practice of the FGM that occurs in the transcultural moment.
The active participation of women in the awareness and sensitization of the negative aspects of the GFM (for migrant women and health professionals).
The integration of women into collaborative networks and associations to develop strategies and actions for the eradication of the GFM.
4.4 Limits of the study:
Problems in identifying groups of migrant women who have experienced the GFM.
Need for collaboration with intermediary institutions (ngos, social services, associations) to connect with groups of migrant women who have experienced the GFM
The difficulties of communicating about the GFM with migrant women who are in a multicultural moment.
The GFM is a very complex issue that provokes fear and suspicion among migrant women who are in a multicultural moment.
Women who are in a transcultural situation have suffered the consequences of their change of attitude towards the GGM: rejection by their husbands, disaffection from their families, marginalization in their communities of origin, uprooting from migrant groups in a multicultural situation.
Point 5: The conclusion need modification.
Response 5: The conclusions have been modified
Point 6: References need update.
Response 6; References have been updated

Reviewer 3 Report
This is an important qualitative study of 7 women who have had experience of female genital mutilation, who have emigrated from their original countries and are now advocating against the practice.
The English is very patchy and, at times, hard to understand.
It is not clear whether the women consented to have their statements published nor whether the study was approved by an ethics committee. It is not clear whether the particulars of each woman are factual or disguised. To avoid identification, it might be better if they were disguised.
It is not clear whether the statements of these 7 women have already been published within the original study of 18 women. If this is so, what exactly is the contribution of this manuscript?
The manuscript uses many terms that are not generally understood in the health field: "cultural moment," "habitus," "dialectic" and many more. Are they necessary?
If these 7 women were chosen because they are advocating against female cutting, would it make sense to compare their responses to those of the 11 women who were not advocating? Otherwise, why focus on them?
Author Response
Point 1 The English is very patchy and, at times, hard to understand.
Response 1: The manuscript was sent to English Editing service of IJERPH for english language review
Point 2: : It is not clear whether the women consented to have their statements published nor whether the study was approved by an ethics committee. It is not clear whether the particulars of each woman are factual or disguised. To avoid identification, it might be better if they were disguised.
Response 2:
2.1 Ethics declarations:The Ethics Committee of the University of Alicante (Spain) approved
this project on January 2018/Number: UA-2017-12-15.This study is conformed to the ethical
principles set out in the Declaration of Helsinki. All the participating individuals signed their
informed consent to participation, after having been explained their rights as citizens, the characteristics and objectives of the study, and their guaranteed anonymity.
2.2 Women in transcutural situation in this research, contribute from different associations and in a public way to the eradication of the GFM. In this research, however, the anonymity of all participants has been preserved.
Point 3: It is not clear whether the statements of these 7 women have already been published within the original study of 18 women. If this is so, what exactly is the contribution of this manuscript?
Response 2: A previous study with all women is not published (18)
Point 3: The manuscript uses many terms that are not generally understood in the health field: "cultural moment," "habitus," "dialectic" and many more. Are they necessary?
Response 3:
All these concepts are interrelated and relevant to assess complex phenomena whose dynamics depends simultaneously on several factors: ideas, beliefs, myths, family situation, migration, culture, scenarios, power relations, cultural level, social class and economy, social
representation of the body and of women…
Cultural moments facilitate the assessment of the level of women's acculturation with respect
to the practice of the GFM. Through an analysis of migrant women's communication with
women from other cultures, we identify the impact of multicultural, intercultural and
transcultural factors in the maintenance, questioning or abandonment of the FGM. In short, it
is a question of evaluating the level of communication between migrant women and women
from other cultures.
Bordieu's habitus clarifies the incidence of women's socialized subjectivity in the
maintenance of certain beliefs, traditions and practices (such as the FGM). The concept of
logical coformism reveals the weight of social fact as an essential element of the coercive
system that maintains order, cohesion and social hierarchy. Gender constitutes one of the
most hierarchical and fundamental categories of power relationships. The SDMC facilitates
the relational analysis of the ideals (beliefs, stories, traditions, myths), personal
characteristics and scenarios of the people directly involved in the practice of GFM (girls,
mothers, grandmothers, cutters and community leaders). It is very pertinent to value the
cultural moments and the evolution of the situation of the emigrant women from the
perspective of the logical conformism and the habitus
Point 4: If these 7 women were chosen because they are advocating against female cutting, would it make sense to compare their responses to those of the 11 women who were not advocating? Otherwise, why focus on them?
Response 4: The purpose of the study is to assess the process by which women who have experienced the GFM (and who were in favour of it in the past, before they migrated to other
countries) begin to question the GFM and the beliefs and myths behind it. The whole study:
theoretical approach centered on the socio-critical paradigm, cultural moments, logical
conformism, habitus, ecteria; is oriented to that procedural assessment that constitutes the
change of mentality and behavior lived by migrant women with respect to this practice.
In a future it would be interesting to deepen in the situation of multiculturalism that maintains
the emigrants women in the same practices and beliefs of their origin communities. The main
problem is the difficulty of communicating with migrant women who remain faithful to the
beliefs and practices of their communities of origin and who are suspicious of any dialogue
on the subject. In any case the study should have a very different conception and design using
communication strategies adapted to the situation
